# Low Resource Competition, Availability of Nutrients and Water Level Fluctuations Facilitate Invasions of Australian Swamp Stonecrop (*Crassula helmsii*)

Hein H. van Kleef [1,2,3,4,*], Janneke M. M. van der Loop [1,2,3] and Laura S. van Veenhuisen [1,2,3]

1    Bargerveen Foundation, Toernooiveld 1, 6525 ED Nijmegen, The Netherlands; j.vanderloop@science.ru.nl (J.M.M.v.d.L.)
2    Radboud Institute for Biological and Environmental Sciences (RIBES), Radboud University, P.O. Box 9010, 6500 GL Nijmegen, The Netherlands
3    Netherlands Expertise Centre Exotics (NEC-E), Toernooiveld 1, 6525 ED Nijmegen, The Netherlands
4    Department of Environmental and Life Sciences, Biology, Karlstad University, Universitetsgatan 2, 651 88 Karlstad, Sweden
*    Correspondence: h.vankleef@science.ru.nl

**Abstract:** Australian swamp stonecrop (*Crassula helmsii* (Kirk) Cockayne) is invasive in Western Europe. Its small size and high potential for regeneration make it difficult to eliminate. Short-term experiments have demonstrated that the growth of *C. helmsii* depends on nutrient availability and resource competition. In order to confirm those mechanisms in the field, we studied the abundance of *C. helmsii* in Northern Europe over a longer period of time in relation to nutrient availability and co-occurring plant communities and plant species. *C. helmsii* impacted native species mainly by limiting their abundance. The native plant species present indicated that previous or periodic elevated nutrient availability were likely responsible for the proliferation of *C. helmsii*. When growing in submerged conditions, the dominance of *C. helmsii* depended on a high availability of $CO_2$. A series of exceptionally dry summers allowed *C. helmsii* to increase in cover due to weakened biotic resistance and a loss of carbon limitation. Only *Littorella uniflora* (L.) Asch. and *Juncus effusus* L. were able to remain dominant and continue to provide biotic resistance. Based on our findings, minimizing nutrient (C and N) availability and optimizing hydrology provides native species with stable growth conditions. This optimizes resource competition and may prevent the proliferation of *C. helmsii*.

**Keywords:** biotic resistance; invasive species; nutrient limitation; weed

## 1. Introduction

Australian swamp stonecrop, *Crassula helmsii* (Kirk) Cockayne, is considered invasive in Western Europe and in the Southeastern United States. This amphibious species is native to Australia and New Zealand but has been imported to many places around the world as an ornamental species for aquaria and garden ponds. *C. helmsii* regularly invades a wide variety of artificial and natural habitats, resulting in dense monospecific stands which can outcompete native flora and change natural ecosystems fundamentally, including their physical and chemical conditions [1–4]. In Northwestern Europe, this species has been rapidly colonizing sites where restoration measures within the framework of the EU Birds and Habitats Directives or EU Water Framework Directive have been taken. There, it prevents the biological recovery of endangered habitats such as humid dune slacks (EU Habitats Directive Annex I Habitat H2190) and oligotrophic waters containing very few minerals (Annex I Habitats H3110 and H3120) [5].

With the exception of small, isolated, terrestrial infestations, this species is nearly impossible to eliminate using chemical, mechanical and physical methods [3,4,6,7], because only a few remaining seeds and/or small fragments are required for the species to

recover from population control measures [8]. To make matters worse, methods to prevent zoochorous and anthropogenic recolonization are lacking [9].

Another way to approach the management of problematic invasive species is to increase the ability of the local environment and resident community to resist invasion [10,11]. *C. helmsii* is not always dominant and does not often outcompete other plant species when they are abundantly present, but it is occasionally also found to have low cover (pers. obs. the authors). This indicates that certain environmental filters determine the invasion success of *C. helmsii* and may be manipulated in order to control the abundance of the species. This requires a solid knowledge base of the relationship between the species' dominance and controlling environmental parameters.

Previously, the range of conditions under which the species occurs has been described extensively [1,4]. *C. helmsii* occurs in various habitats. These habitats vary from slow-flowing and standing waters to lake shores where the species grows submerged as well as exposed on banks above the water level, tolerating water level fluctuations [12]. *C. helmsii* has been found growing on clay, sand, gravel, silt and organic soils [1]. In its native range, the plant appears in waters with low conductivity as well as in brackish waters, and it occurs under varying nutrient levels [12].

The abundance of *C. helmsii* under field conditions is poorly studied, but may vary considerably depending on local conditions [13]. To our knowledge, the only work on this topic has been performed by Brunet [14], who studied a large number of sites in England and found that the occurrence and cover of the species increased at a pH above 5 and a conductivity above 125 μS. Furthermore, the species appears to be limited by shading and competition with large plants [1,12], as well as prolonged desiccation [12]. Short-term experiments have also provided evidence that the abundance of *C. helmsii* increases with nutrient availability (N and P) [8,15,16]. In addition, field and lab experiments revealed that native competing species limit the ability of *C. helmsii* to access below-ground nutrients, reducing its growth [15–17].

The abovementioned studies provide a comprehensive description of the species' niche. Experiments provide the best environmental factors for controlling the invasiveness of *C. helmsii*, i.e., limiting nutrient availability and promoting interspecific competition. Whether these factors are also effective in determining the dominance of *C. helmsii* outside of controlled experimental designs and under spatially and temporally varying field conditions remains to be seen. Therefore, we conducted a field study to answer the following questions:

- Is the cover of *C. helmsii* limited by nutrient availability? And which nutrients are important?
- Is the cover of *C. helmsii* limited by interspecific competition? If so, which native species are involved?
- How does the cover of *C. helmsii* develop over time, and what is its impact on native flora?

We studied the dominance of *C. helmsii* at different locations within its non-native range in Northern Europe in relation to biotic and abiotic site properties. Because *C. helmsii* is an amphibious species, we distinguished between aquatic and terrestrial growth sites. This study was repeated in a subset of locations to assess the temporal stability of communities and competing species.

## 2. Materials and Methods

### 2.1. Selection of Study Sites

Forty-eight locations with observations of *C. helmsii* were selected (Supporting Information Table S1) from the Dutch National Flora and Fauna Database (NDFF), which contains distribution data of plants and animals in the Netherlands. Only natural sites were selected, whereas anthropogenically impacted sites in agricultural landscapes and urban areas were excluded from the study, although influences from former or nearby agricultural activities could not be ruled out. These criteria were also used to select twenty-three sites

with *C. helmsii* in the south of England. In total, thirty-seven sites were located on the moist banks of surface waters, whereas thirty-four sites were aquatic.

## 2.2. Environmental Data on Water and Soil Chemistry, Precipitation, and Management

A surface water sample was taken at every location where *C. helmsii* was found growing in the water. Water samples were collected in iodated polyethylene bottles. The pH, alkalinity, and Total Inorganic Carbon (TIC) were measured within 24 h after collection. The pH was measured with a standard combined glass Ag/AgCl pH electrode (Orion Research, Beverly, CA, USA) connected to a pH meter (Tim800; Radiometer analytical, Lyon, France). Alkalinity was measured through the titration of 50 mL of surface water with 0.01 mmol·L$^{-1}$ of HCl down to pH 4.2 using an auto burette (ABU901, Radiometer, Lyon, France). TIC was determined using infrared gas chromatography (IRGA; ABB Advance Optima, Zurich, Switzerland). Based on pH and TIC concentrations, $CO_2$ and $HCO_3^-$ contents of the water were calculated according to the method by Stumm and Morgan [18]. After filtering (Whatman GF/C filter) and adding 1 mg of citric acid per 25 mL of water, samples were stored at $-20\ ^\circ$C until further analysis of $NH_4^+$, $NO_3^-$ and P.

At each location, five subsamples were taken from the top 5 cm of soil using a gouge auger. The subsamples were pooled to correct for soil heterogeneity. When taking soil samples under water, particular care was taken to keep the samples intact. Soil chemical characteristics were determined using three different methods. Fresh soil (17.5 g) was mixed with 50 mL 0.2 mol·L$^{-1}$ NaCl solution for determining $NH_4^+$. Another sample of 17.5 g fresh soil was mixed with 50 mL of water to determine $NO_3^-$. Remaining soil was dried at 70 $^\circ$C until a constant weight was reached to determine total P.

An auto-analyzer 3 system (Bran and Lubbe, Norderstedt, Germany) was used to colorimetrically measure concentrations of $NO_3^-$ and $NH_4^+$ in surface water and soil extracts using hydrazine sulphate [18] and salicylate [19], respectively. An inductively coupled plasma spectrometer (ICP-OES icap 6000; Thermo Fischer Scientific, Waltham, MA, USA) was used to measure the concentration of P in surface water. Total P of soil samples was determined by digesting 200 mg of dried and homogenized soil sample in 4 mL 65% $HNO_3^-$ and 1 mL 30% $H_2O_2$ in Teflon containers using an Ethos D microwave (Milestone, Sorisole Lombardy, Italy). The digests were diluted in 50 mL of ultrapure mili-Q water (Milipore Corp., Burlington, MA, USA) and analyzed by inductively coupled plasma emission spectrometry (ICP-OES icap 6000; Thermo Fischer Scientific, Waltham, MA, USA).

Data on cumulative precipitation shortage on August 31 were obtained for the period of 1971–2020 from the Royal Netherlands Meteorological Institute (KNMI) [19]. They are Which is based on data collected at thirteen reference weather stations in the Netherlands located in De Bilt, De Kooy, Groningen, Heerde, Hoofddorp, Hoorn, Kerkwerve, Oudenbosch, Roermond, Ter Apel, West-Terschelling, Westdorpe and Winterswijk.

Data on former use and management of the sites were obtained from the site owners or managers.

## 2.3. Vegetation Survey and Biomass Measurement

At each site, a homogeneous vegetation was chosen in which *C. helmsii* was present. All plant species were identified within a plot of 4 by 4 m, and their covers were estimated in percentages. The surveys were performed by two observers who had trained and worked together for several years. If we were unable to census a 4 × 4 m plot, a plot was chosen with a different shape but also with a surface area of 16 m$^2$.

Within each 16 m$^2$, a subplot of 0.5 × 0.5 m was randomly placed in which all aboveground biomass of *C. helmsii* was clipped and collected. When cover and biomass of *C. helmsii* were especially large, a smaller sample of 0.25 × 0.5 m was taken. Biomass samples were dried in paper bags for 72 h at 70 $^\circ$C, and their contents were weighed and converted to gram dry weight per m$^2$.

Vegetation surveys were conducted in September 2016 in the Netherlands and in October 2019 in England, at which time water and soil samples were also collected. In

September 2020, most Dutch sites were re-visited for a new vegetation survey. Two sites were excluded from the 2020 survey because they had been subject to intensive nature management and no longer represented natural succession trajectories.

*2.4. Statistical Analysis*

Measured nutrient levels may fluctuate over time, for example, due to intake and accumulation in plants. Another way to assess nutrient status is through inference from the environmental preferences of the plant species present. This was carried out using the Ellenberg indicator value for nitrogen [20], which, according to Hill et al. [21], is a general indicator of soil fertility. Each plant was assigned an indicator value ranging from 1 (extremely infertile sites) to 9 (extremely rich situations). Data were available for 95% of the native vascular plant species [20,21]. No indicator values were available for bryophytes, algae and non-native species. The community indicator value was therefore calculated using only data on the presence and abundance of species, for which indicator values were available. Mean Ellenberg indicator value for nitrogen of each community was calculated as

$$\text{Mean EV} = \sum \frac{(p_i EV_i)}{p_i} \tag{1}$$

where $p_i$ is the cover (%) of a species and EV is the Ellenberg indicator value for nitrogen.

A measure of the relative biomass was calculated, describing whether the measured biomass of *C. helmsii* per unit area on a site was higher or lower than expected based on its cover. Therefore, a curve was fitted to describe the relationship between log-transformed observed biomass (M) and *C. helmsii* cover (A), which yielded the following equation:

$$\text{Log}(M+1) = -0.0002A^2 + 0.0442A + 0.4601 \tag{2}$$

Using Equation (2), the expected biomass on a site (M′) could be calculated as follows:

$$M' = 10^{(-0.002A^2 + 0.0442A + 0.4601)} - 1 \tag{3}$$

Relative biomass (RB) was calculated as the difference between expected and observed biomass and divided by expected biomass, because the ability to grow depends on the number of plants present:

$$RB = \frac{M' - M}{M'} \tag{4}$$

Relevés from the 2016 and 2020 Dutch surveys were used to describe dynamics of communities with *C. helmsii*. For this, sites with comparable species composition were identified using a two-way indicator species analysis (Twinspan) [22]. Abundance data were divided into six cutting levels of 0–5, 6–10, 11–25, 25–50, 51–75 and 76–100% cover, respectively. Community dynamics were also described using three metrics: change in species richness, richness-based species exchange ratio ($SER_r$, Equation (5)), and abundance-based species exchange ratio ($SER_a$, Equation (6)). $SER_r$ was calculated following Hillebrand et al. [23] as follows:

$$SER_r = \frac{S_{imm} + S_{ext}}{S_{tot}} \tag{5}$$

where $S_{imm}$ is the number of species newly recorded, $S_{ext}$ is the number of species no longer recorded, and $S_{tot}$ is the total number of species of both years combined. $SER_a$ is comparable to $SER_r$. Instead of describing turnover within a community using species numbers, $SER_a$ uses species proportional abundances. $SER_a$ was calculated following Equation (6). See Hillebrand et al. [23] for an explanation of the math.

$$SER_a = \frac{\sum_i (p_i - p_i')^2}{\sum p_i^2 + \sum p_i'^2 - \sum p_i p_i'} \tag{6}$$

For the statistical analysis of *C. helmsii* cover, we distinguished sites where *C. helmsii* had a cover of <50% and sites where the species was dominant with a cover >50%. We also grouped sites by indication of low to intermediate fertility (Mean EV < 4) or intermediate to high fertility (Mean EV ≥ 4). Shapiro–Wilk tests for normality and F tests for equal variance were performed on species cover and richness data, *C. helmsii* biomass, mean Ellenberg values, $SER_r$, $SER_a$, and water and soil chemistry. Since assumptions for normality and equal variance were not met, comparisons between groups were performed with the Wilcoxon rank-sum test or Wilcoxon signed-rank test. Kendall's Tau was used to test for non-parametric correlations. Data were tested and visualized using the statistical program R version 4.1.0. [24] using the tidyverse package [25] for data manipulation and the ggplot2 [26] and ggsignif packages [27] for data visualization.

### 3. Results

The study sites differed markedly in the cover and biomass of *C. helmsii*. The biomass varied between 0.004 and 1971 g of dry weight per $m^2$ and increased with cover (Figure 1). The mean biomass did not differ between the aquatic and terrestrial sites ($W(69) = 613$, $p = 0.859$). The cover of *C. helmsii* was not evenly distributed, but it was either low or high. Only a few relevés were characterized by intermediate cover (Figure 1).

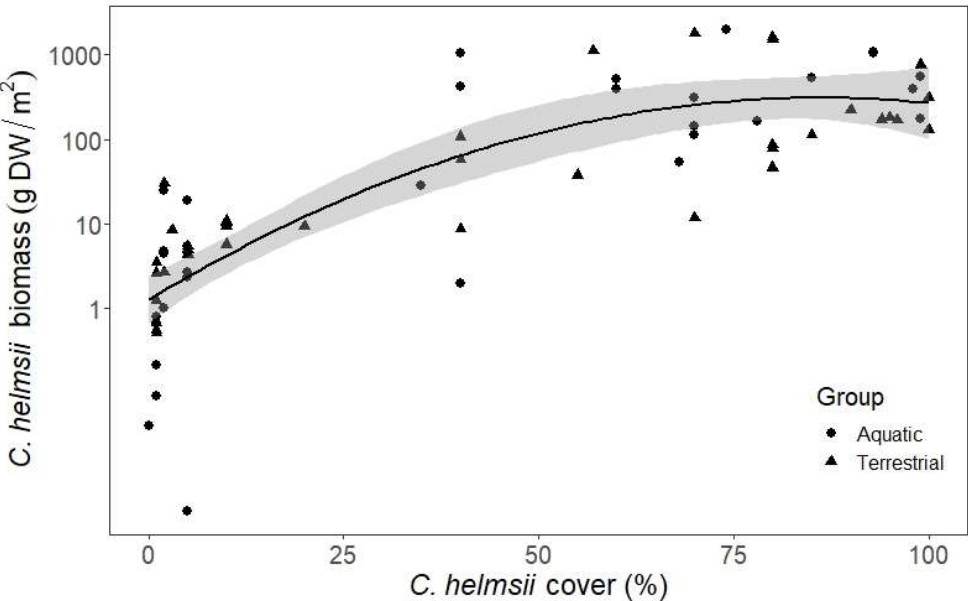

**Figure 1.** Cover (%) and biomass (grams dry weight per $m^2$, note: logarithmic scale on the y-axis) of *Crassula helmsii* in study sites (N = 71). Line describes relationship between *C. helmsii* cover and biomass (as in Equation (3)) with 95% confidence intervals.

The cover of the species, other than *C. helmsii*, was higher at sites where *C. helmsii* cover was low (Table 1). Species richness was lower with a high *C. helmsii* cover in the terrestrial sites but not in the aquatic sites (Table 1). The mean Ellenberg value for N was the highest in the terrestrial sites where the cover of *C. helmsii* was high (Table 1). However, this apparent nutrient richness could not be confirmed with measurements, as the nitrogen and phosphorous concentrations in water and soil in general were low and did not differ between sites with low or high covers of *C. helmsii* (Table 1). The organic content of soil differed between countries ($W(69) = 41$, $p < 0.001$). The Dutch sites were predominantly low in organic matter (mean 3 ± 3%), whereas on the English sites, the organic matter content was relatively high (mean 38 ± 4%). The organic soil content did not differ significantly between the sites with low or high covers of *C. helmsii* (Table 1). The carbon dioxide concentrations in water bodies with a high cover of *C. helmsii* were higher than in sites where the cover was low (Table 1). There was a small but significant difference in the pH

of water bodies differing in cover (Table 1). In general, the study sites were slightly acidic to neutral and poorly buffered, with an alkalinity of less than 1.0 meq·L$^{-1}$ (Table 1). The exceptions were the Groot Meer site, which is supplied by alkaline ground water, and most of the English sites, which are situated on limestone bedrock. At these sites, the pH varied between 6.9 and 9.0, and the alkalinity ranged between 0.3 and 3.3 meq·L$^{-1}$. The alkalinity did not differ significantly between sites with low and high covers of *C. helmsii* (Table 1).

**Table 1.** Water and soil chemistry and plant community characteristics of sites with low (<50%) and high (≥50%) cover of *Crassula helmsii*. Presented data are medians (with 5 and 95% intervals) and Wilcoxon rank-sum exact tests (*W*, *p* and *df* values).

| | Subset: | Aquatic Sites | | | | | Terrestrial Sites | | | | |
|---|---|---|---|---|---|---|---|---|---|---|---|
| | *C. helmsii* Cover: | Low (N = 20) | High (N = 14) | *W* | *p* | *df* | Low (N = 20) | High (N = 17) | *W* | *p* | *df* |
| **Community** | | | | | | | | | | | |
| | Cover other species (%) | 79.3 (11.0, 99.0) | 21.3 (1.0, 35.3) | 24 | **<0.001** | 32 | 90.0 (48.5, 99.0) | 11.0 (0.0, 24.2) | 3 | **<0.001** | 35 |
| | Species richness | 5.0 (2.0, 13.0) | 6.0 (2.3, 12.4) | 147 | 0.819 | 32 | 9.5 (3.9, 16.3) | 4.0 (1.0, 12.4) | 95 | **0.023** | 35 |
| | Mean Ellenberg value for N | 2.4 (2.0, 7.0) | 4.5 (2.0, 6.2) | 154 | 0.634 | 32 | 2.1 (1.8, 5.7) | 3.82 (1.4, 7.2) | 237 | **0.043** | 35 |
| **Surface water chemistry** | | | | | | | | | | | |
| | pH | 7.3 (5.6, 8.5) | 6.7 (5.6, 7.3) | 75 | **0.023** | 32 | | | | | |
| | Alkalinity (meq/L) | 0.8 (0.1, 3.0) | 0.5 (0.1, 3.1) | 140 | 1.000 | 32 | | | | | |
| | CO$_2$ (μmol/L) | 51.6 (1.6, 287) | 225.9 (48.4, 940.5) | 225 | **0.003** | 32 | | | | | |
| | Total N (μmol/L) | 13.3 (7.7, 79.3) | 11.1 (2.9, 77.2) | 115 | 0.396 | 32 | | | | | |
| | Total P (μmol/L) | 0.7 (0.0, 7.1) | 0.3 (0.0, 7.0) | 135 | 0.872 | 32 | | | | | |
| **Soil chemistry** | | | | | | | | | | | |
| | Soil organic matter (%) | 6.5 (0.8, 38.8) | 5.3 (0.9, 37.4) | 106 | 0.950 | 28 | 2.1 (0.8, 41.6) | 10.2 (0.7, 43.4) | 189 | 0.397 | 34 |
| | Total N (μmol/g DW) | 0.22 (0.04, 0.66) | 0.19 (0.02, 1.70) | 100 | 0.755 | 28 | 0.19 (0.04, 2.81) | 0.21 (0.05, 2.86) | 179 | 0.798 | 35 |
| | Total P (μmol/g DW) | 2.8 (0.8, 16.0) | 10.9 (1.1, 20.0) | 109 | 0.487 | 28 | 1.8 (0.9, 18.5) | 2.5 (1.0, 28.2) | 197 | 0.153 | 35 |

The *C. helmsii* biomass was higher than expected when the mean Ellenberg values for nitrogen were high in the aquatic study sites, but not in the terrestrial sites (Figure 2).

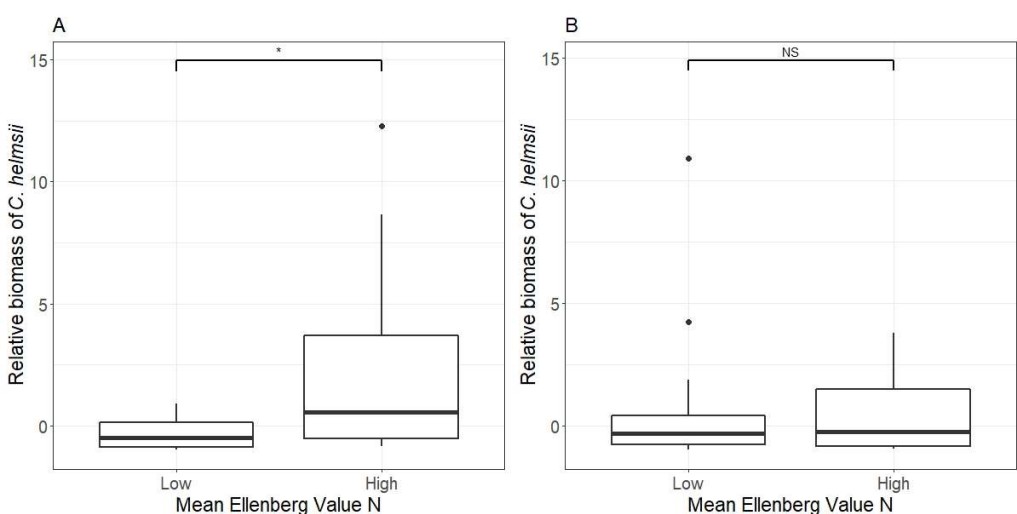

**Figure 2.** Relation between relative biomass of *Crassula helmsii* and nutrient availability of the ecosystem, as indicated by the identity and cover of the species present. (**A**) Aquatic sites; (**B**) terrestrial sites. Low: Mean Ellenberg value of N < 4; high: Mean Ellenberg value of N ≥ 4. * *W*(32) = 215; *p* = 0.014 Wilcoxon rank-sum exact test. NS: *W*(35) = 134; *p* = 0.987 Wilcoxon rank-sum exact test.

A cluster analysis distinguished six vegetation communities (Table 2). These represent a complete moisture gradient, from permanently to only occasionally inundated. The aquatic and amphibious communities (I, II, III and IV) are characterized by lower species richness than the terrestrial communities (V and VI). Community I is a species-poor community, consisting of aquatic and amphibious species, with a dominance of *Elatine hexandra* (Lapierre) DC. In community II, *E. hexandra* is still present, but no longer abundant. This more diverse community is dominated by *Pilularia globulifera* L., which thrives on recently exposed soils. Here, amphibious species, such as *Baldellia ranunculoides* subsp. *ranunculoides* (L.) Parl., are frequently found. Community III is also aquatic or found on wet shores

that are regularly inundated. It is characterized by a high presence of *Juncus bulbosus* L. and *Hydrocotyle vulgaris* L., and occasionally dominated of *Littorella uniflora* (L.) Asch. and *Warnstorfia fluitans* (Hedw.) Loeske. Communities IV and V contain sites with the highest covers of *C. helmsii*. Of these two, IV is the wettest community with the occasional presence of aquatic (*E. hexandra*, *Potamogeton polygonifolius* Pourr.) and amphibious species, such as *P. globulifera*. Community V has a high presence of species growing on moist soils (i.e., *Eleocharis multicaulis* (Sm.) Desv., *H. vulgaris*), but also many species that only moderately tolerate inundation (i.e., *Agrostis canina* L., *Leontodon saxatilis* Lam., *Gnaphalium luteoalbum* (L.) Hilliard & B.L.Burtt, *Erigeron canadensis* L.). Community VI includes shores that become inundated infrequently, only during periods of high precipitation. This is reflected in the frequent occurrence of terrestrial species, such as *Lotus pedunculatus* Cav., *A. canina*, *L. saxatilis*, *Ranunculus flammula* L. and *Lycopus europaeus* L. This community is characterized by a high presence and cover of species that are often found on the shores of nutrient-poor shallow soft water lakes, such as *E. multicaulis*, *H. vulgaris* and *Hypericum elodes* L. Occasionally, species that prefer more nutrient-rich conditions are present in community VI, such as *Juncus effusus* L. and *Phragmites australis* (Cav.) Trin. ex Steud.

**Table 2.** A summary of a synoptic table of identified plant communities based on a TWINSPAN analysis using surveys from the Dutch sites in 2016 and 2020. Only species with a presence of 0.25 or higher in one or more of the communities are listed. The numbers give the fraction of relevés where a species was recorded. Fractions equal to or higher than 0.25 are highlighted in bold to distinguish characteristic species.

| | Community | | | | | |
|---|---|---|---|---|---|---|
| | I | II | III | IV | V | VI |
| Number of relevés: | 7 | 6 | 10 | 28 | 15 | 26 |
| Mean species richness, excluding *Crassula helmsii* | 3.6 | 6 | 6 | 4.3 | 11.2 | 12.9 |
| Mean *C. helmsii* cover | 6.1% | 3.5% | 3.8% | 82.7% | 84.4% | 5.7% |
| *Eleocharis acicularis* | **0.29** | | | 0.11 | | |
| *Chara* sp. | **0.29** | | 0.20 | 0.04 | | |
| *Pilularia globulifera* | **0.29** | **0.83** | | 0.18 | 0.13 | 0.08 |
| *Elatine hexandra* | **1.00** | **1.00** | 0.20 | 0.18 | | 0.04 |
| *Alisma plantago-aquatica* | **0.29** | **0.33** | 0.20 | 0.07 | 0.13 | |
| *Potamogeton polygonifolius* | | **0.33** | | 0.04 | | |
| *Baldellia ranunculoides* subsp. *Ranunculoides* | | **0.83** | | 0.07 | | |
| *Lythrum portula* | 0.14 | **0.33** | 0.10 | 0.04 | 0.20 | 0.23 |
| *Crassula helmsii* | **1.00** | **1.00** | **1.00** | **1.00** | **1.00** | **1.00** |
| *Juncus bulbosus* | **0.57** | **0.33** | **0.80** | **0.25** | **0.47** | **0.35** |
| *Lysimachia vulgaris* | | **0.33** | **0.30** | 0.11 | 0.20 | **0.38** |
| *Salix* sp. | | 0.17 | 0.10 | 0.11 | **0.33** | **0.54** |
| *Eleocharis multicaulis* | | 0.17 | 0.10 | **0.29** | **0.60** | **0.65** |
| *Hydrocotyle vulgaris* | | | | **0.50** | **0.29** | **0.40** | **0.88** |
| *Hypericum elodes* | | | | **0.30** | 0.25 | 0.13 | **0.69** |
| *Bidens frondosa* | | | 0.10 | 0.21 | **0.73** | **0.27** |

**Table 2.** *Cont.*

| | | | Community | | |
|---|---|---|---|---|---|
| *Gnaphalium luteoalbum* | | | | **0.47** | 0.08 |
| *Mentha aquatica* | | 0.10 | 0.04 | **0.73** | 0.12 |
| *Erigeron canadensis* | | | 0.04 | **0.40** | 0.08 |
| *Digitaria ischaemum* | | | | **0.27** | |
| *Epilobium hirsutum* | | | 0.04 | **0.27** | 0.15 |
| *Betula pubescens* | | | | **0.33** | 0.23 |
| *Carex oederi* | | | | **0.27** | 0.23 |
| *Lotus pedunculatus* | | | | **0.27** | **0.42** |
| *Agrostis canina* | | 0.10 | 0.04 | **0.40** | **0.62** |
| *Leontodon saxatilis* | | | | **0.47** | 0.27 |
| *Ranunculus flammula* | | 0.10 | 0.04 | **0.40** | **0.35** |
| *Lycopus europaeus* | 0.17 | | 0.18 | **0.80** | **0.81** |
| *Juncus effusus* | | | 0.18 | 0.13 | **0.42** |
| *Phragmites australis* | | 0.20 | 0.18 | 0.07 | **0.31** |
| *Galium palustre* | | | | | **0.65** |
| *Moss* sp. | | | 0.14 | | **0.31** |

From 2016 through 2020, the *C. helmsii* cover increased by more than 20% at 13 sites and decreased by more than 20% at another 13 sites. At 20 sites, the change in cover was less than 20%. From 2016 through 2020, the mean species richness in the study sites increased significantly from 6.7 ± 0.68 to 9.78 ± 0.74 (Paired Wilcoxon test, $V(44) = 124$, $p < 0.001$). The mean species- and abundance-based species exchange ratios were high, with values of 0.76 and 0.77, respectively, indicating a high turnover in species identity as well as abundance. There was no relation between change in species richness and change in the cover of *C. helmsii* (Figure 3A). High values of $SER_r$ and $SER_a$ were recorded independent of the change in the *C. helmsii* cover (Figure 3B,C). When the change in the *C. helmsii* cover was small, SER. and $SER_a$ were low at a few sites.

In the period of 1971–2020, the mean cumulative precipitation deficit on August 31 in the Netherlands was 120 mm. 2016 was a relatively wet year, with a cumulative precipitation deficit of 72 mm on August 31. The year of 2017 was a relatively normal year with respect to precipitation, with a cumulative precipitation deficit of 140 mm on August 31. The following three years, however, were dry and had cumulative precipitation deficits of 287, 206 and 221 mm, respectively, on August, 31. In the period of 2016–2020, wet communities I and II almost disappeared (Table 3). In most of these sites, the plant community changed to communities IV and V and the cover of *C. helmsii* increased. Community III was more stable. In half of the sites that classified as community III in 2016, the vegetation was also classified as community III in 2020. A third of the sites that were classified as community III in 2016 changed to community IV in 2020, where the cover of *C. helmsii* was higher. Communities IV and V, where *C. helmsii* already dominated, were also impacted by the drought, as 9 out of 20 sites developed towards the dryer community VI, resulting in a lower cover of *C. helmsii*. Almost a third of the sites with wet community IV dominated by *C. helmsii* shifted towards the dryer community V, still harboring a high cover of *C. helmsii*. Most sites belonging to community VI in 2016 remained in the same community in 2020. However, they were not completely stable, because in the terrestrial sites, the dominant species in 2016 were often replaced by other species in 2020 (Table 4).

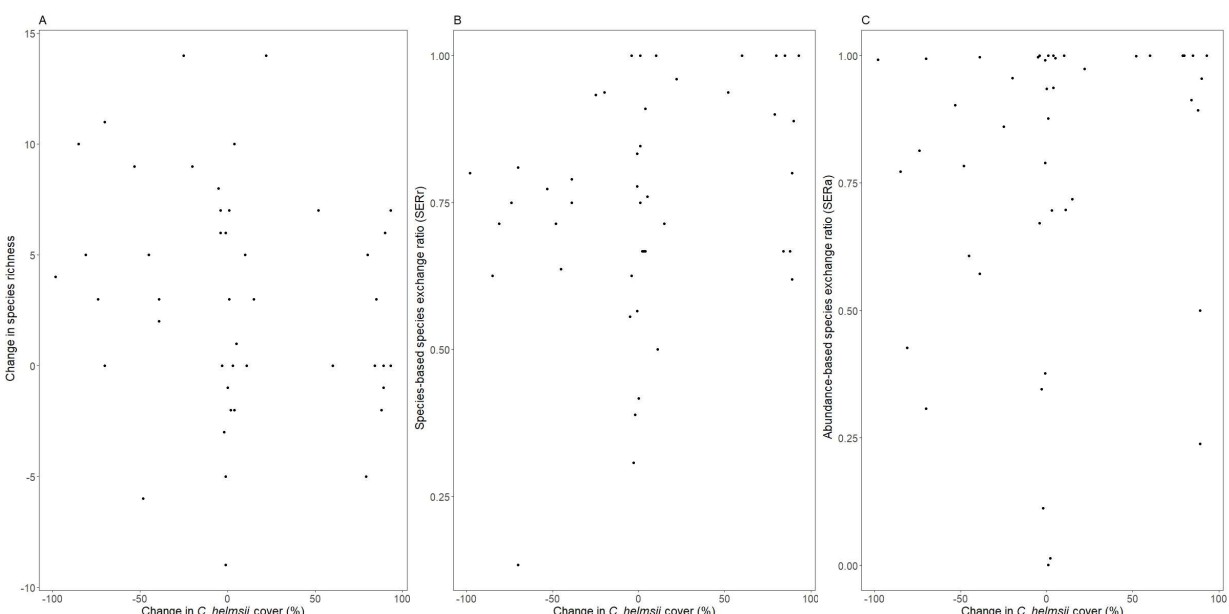

**Figure 3.** Bivariate plots between change in species richness (**A**), species- (**B**) and abundance-based turnover (**C**) and change in cover of *Crassula helmsii*.

**Table 3.** Changes in the plant communities in the Dutch study sites. The rows describe to which community the study sites belonged in 2016. The columns describe to which community these sites belonged in 2020. The numbers are the number of sites that classified for the communities in 2016 and 2020, e.g., of the six sites classified as community I (a species poor aquatic community with *Elatine hexandra* dominance) in 2016, five were classified as community IV and one was classified as community V in 2020.

|  |  | 2020 | | | | | | |  |
|  |  | I | II | III | IV | V | VI | *N* |  |
|  | I |  |  |  | 5 | 1 |  | 6 | Species-poor aquatic community with *Elatine hexandra* dominance |
|  | II | 1 |  |  | 1 | 3 | 1 | 6 | Aquatic or recently exposed shores with dominance of *Pillularia globulifera* |
| 2016 | III |  |  | 3 | 2 |  | 1 | 6 | Aquatic or recently exposed with dominance of *Juncus bulbosus* |
|  | IV |  |  | 1 | 3 | 6 | 7 | 17 | Mostly species-poor aquatic communities dominated by *Crassula helmsii* |
|  | V |  |  |  |  | 1 | 2 | 3 | Moist exposed shores dominated by *Crassula helmsii* |
|  | VI |  |  |  |  | 1 | 7 | 8 | Dry species-rich shores |
|  | *N* | 1 | 0 | 4 | 11 | 12 | 18 | 46 |  |

The change in *C. helmsii* cover was not correlated with the total nitrogen or phosphorus concentrations in the soil nor to nutrient availability, as indicated by the mean Ellenberg value for N (Kendall's Tau, N = 46, $p > 0.05$). From 2016 through 2020, the expansion of *C. helmsii* mainly took place at aquatic sites, and 8 out of 12 sites with a low cover in 2016 had a high cover of *C. helmsii* in 2020 (Table 4). At terrestrial sites, the cover of *C. helmsii* increased in only 4 out of 16 sites with a low cover in 2016. When no dominant species were present, sites with a low cover of *C. helmsii* all became heavily invaded. Three years with dry summers caused many dominant species to be replaced either by *C. helmsii* or by other native species. Species that were dominant in 2016 on multiple occasions but did not manage to remain abundant in 2020 included the amphibious species *E. hexandria*, *J. bulbosus* and *P. globulifera*. Only a few species had populations that remained stable over time. These included *L. uniflora* at aquatic sites and *J. effusus* at terrestrial sites. At the terrestrial sites, species favoring dry conditions, such as grasses (*Agrostis*, *Calamagrostis*) and mosses, increased.

**Table 4.** Dominant plant species other than *Crassula helmsii* (cover $\geq$ 30%) at sites with different trends in *Crassula helmsii* cover. Increase: *C. helmsii* cover in 2016 <50%, and in 2020, >50%. Stable low: *C. helmsii* cover in both years <50%. Stable high: *C. helmsii* cover in both years >50%. Decline: 2016 >50% and in 2020, <50%. English sites were only surveyed in 2019, and thus, trend in *C. helmsii* abundance was unknown. Numbers give the number of sites where a species was dominant in 2016/2020 or 2019 (column "Unknown").

| Subset: | Aquatic Sites | | | | | |
|---|---|---|---|---|---|---|
| Trend: | Increase | Stable Low | Stable High | Decline | Unknown | |
| *Crassula helmsii* cover in 2016/2019 period: | Low | Low | High | High | Low | High |
| *C. helmsii* cover in 2020: | High | Low | High | Low | | |
| N = | 8 | 4 | 6 | 4 | 8 | 4 |
| No dominant species other than *C. helmsii* | 2/7 | 0/1 | 6/6 | 4/0 | 2 | 4 |
| *Elatine hexandra* | 4/0 | 0/1 | | | | |
| *Warnstorfia fluitans* | 1/0 | | | | | |
| *Juncus bulbosos* | 1/0 | 1/0 | | | | |
| *Littorella uniflora* | | 2/2 | | 0/1 | | |
| *Pilularia globulifera* | | 1/0 | | | | |
| Algae | | 0/1 | | | 4 | |
| *Eleocharis palustris* | | 0/1 | | | | |
| *Agrostis canina* | | | | 0/1 | | |
| *Bidens frondosa* | | | | 0/1 | | |
| *Eleocharis multicaulis* | | | | 0/1 | | |
| *Hypericum elodes* | | | | 0/1 | | |
| *Sphagnum* spec. | | | | 0/1 | | |
| Moss sp. | 0/1 | | | 0/1 | | |
| *Chara vulgaris* | | | | | 1 | |
| *Elodea canadensis* | | | | | 1 | |
| *Elodea nuttallii* | | | | | 2 | |
| *Hydrocharis morsus-ranae* | | | | | 1 | |
| *Hydrocotyle ranunculoides* | | | | | 1 | |
| Subset: | Terrestrial sites | | | | | |
| Trend: | Increase | Stable low | Stable high | Decline | Unknown | |
| *C. helmsii* cover in 2016/2019 period: | Low | Low | High | High | Low | High |
| *C. helmsii* cover in 2020: | High | Low | High | Low | | |
| N = | 4 | 12 | 5 | 5 | 4 | 7 |
| No dominant species other than *C. helmsii* | 2/4 | 0/3 | 5/5 | 5/2 | 0 | 7 |
| *Bidens frondosa* | 1/0 | | | | | |
| *Juncus bulbosos* | 1/0 | 2/0 | | | | |
| *Pilularia globulifera* | | 4/0 | | | | |
| *Eleocharis multicaulis* | | 3/0 | | | | |
| *Juncus effusus* | | 2/2 | | | | |
| *Sphagnum* sp. | | 2/0 | | | | |

**Table 4.** *Cont.*

| Subset: | Aquatic Sites | | | | |
|---|---|---|---|---|---|
| Trend: | Increase | Stable Low | Stable High | Decline | Unknown |
| Algae | | 1/0 | | | |
| *Warnstorfia fluitans* | | 1/0 | | 0/1 | |
| *Agrostis canina* | | 0/4 | | | 1 |
| *Calamagrostis canescens* | | 0/1 | | | |
| *Hydrocotyle vulgaris* | | 0/1 | | | |
| *Radiola linoides* | | 0/1 | | | |
| *Trifolium repens* | | 0/1 | | | |
| Moss sp. | | 0/2 | | 0/1 | |
| *Salix cinerea* | | | | 0/1 | |
| *Hypericum elodes* | | | | | 1 |
| *Isolepis fluitans* | | | | | 1 |
| *Lycopus europaeus* | | | | | 1 |
| *Potentilla anserina* | | | | | 1 |

From 2016 to 2020, the *C. helmsii* cover declined at about half of the sites, which had high cover of the species in 2016, allowing other species to become dominant. At the aquatic sites, these were amphibious species (*L. uniflora*, *E. multicaulis*, *H. elodes* and *Sphagnum* sp.). The terrestrial sites either remained scarcely vegetated or became overgrown with mosses or willow.

The terrestrial study sites in England were similar in plant cover to the Dutch sites, with the presence and high cover of *H. elodes*, *Isolepis fluitans* (L.) R.Br., *A. canina* and *L. europaeus*. However, the aquatic sites in England were dominated by different plant species, such as algae, *Chara vulgaris* L., *Hydrocharis morsus-ranae* L. and the three invasive species *Elodea canadensis* Michx., *E. nuttallii* (Planch.) H.St.John and *Hydrocotyle ranunculoides* L.f. These species favor relatively nutrient-rich growth conditions, which is reflected in a high mean Ellenberg value for N (mean 5.9) for these sites.

## 4. Discussion

### 4.1. Conditions for Optimal Growth

Various studies experimentally demonstrated that elevated nutrient levels contribute to the proliferation of *C. helmsii*, e.g., [15,16]. In this study, the variation in *C. helmsii* could not be linked to the nutrient concentrations measured in soil and water. It is likely that the rapid uptake of nutrients by plants has obscured this relationship (see also [28]), because a relatively high cover of nitrophilous species at sites with a high cover or biomass of *C. helmsii* indicated that at least temporarily elevated nutrient levels contributed to the proliferation of *C. helmsii*. Various sources of nutrients may be involved. Atmospheric nitrogen deposition and droppings of water fowl [16,29] provide a continuous source of nutrients. Many of the Dutch (85%) and English sites (43%) were located on former agricultural fields that, in previous years, had been restored to more natural conditions through topsoil removal (pers. comm. site managers) but may have retained part of the nutrients supplied by the former use. Also, at the other Dutch sites, restoration measures had been taken to restore water bodies from the effects of eutrophication and acidification. In addition, temporary influxes of nutrients may occur. This was the case in Akkerenven, which, in the winter of 2017, received agricultural runoff with high nitrogen (995 $\mu$mol·L$^{-1}$) and carbon concentrations (1280 $\mu$mol·L$^{-1}$), whereas in the summer, the measured concentrations were 11 $\mu$mol·L$^{-1}$ and 87 $\mu$mol·L$^{-1}$, respectively (unpublished data from Bargerveen Foundation). The remaining sites consisted of water bodies (ponds and ditches) located in semi-natural

landscapes dominated by grass production and received nutrient-rich runoff from the managed grasslands [30].

Because *C. helmsii* is only able to assimilate $CO_2$ as a carbon source [28], abundant submerged growth is restricted to water bodies with a pH below 8, at which point inorganic carbon is only available as bicarbonate. The species was not encountered at sites with a pH below 5.6, which is remarkable, as there are many comparable shallow lakes in the Netherlands that are more acidic [31,32]. Brunet [14] also found a lower limit of a pH of 5.7 for the occurrence of *C. helmsii*. According to Van Doorn et al. [28], a low pH may hamper the growth of *C. helmsii* because at a low pH, concentrations of aluminum increase to toxic levels. When the carbon concentration in the water is low, *C. helmsii* plants remain small so they can use carbon that is released from the sediment (pers. obs.). Furthermore, the species possesses Crassulacean acid metabolism (CAM) photosynthesis, allowing it to minimize respiratory carbon loss [33]. Despite *C. helmsii* having these strategies to deal with a limited availability of $CO_2$ in the water, its growth is limited below 100 $\mu mol \cdot L^{-1}$, and an exuberant submersed growth requires the mean concentration of $CO_2$ in the summer to exceed 200 $\mu mol \cdot L^{-1}$ (Table 1). A comparable threshold for proliferation in soft water lakes was found for the elodeids *Myriophyllum alterniflorum* DC. and *Callitriche hamulata* Kütz. ex W.D.J. Koch [34]. Carbon limitation can be lifted by runoff from (former) agriculture pastures, where lime is used to increase crop productivity. The global rise in atmospheric $CO_2$ concentrations may also play a role. In terrestrial ecosystems, it may have increased vegetation productivity by 10–50% over the past 50 years [35]. Little is known about how much the global rise in $CO_2$ contributes to carbon availability in freshwater. However, the consensus is that the carbon availability has increased [36,37], and thus may have contributed to *C. helmsii*'s invasiveness.

Hussner [13] recorded *C. helmsii* on soils with an organic matter content within the range of 3–10%. In the current study, growth of *C. helmsii* did not appear to be affected by the amount of organic matter in the soil, and organic fractions up to 43% were recorded. Despite the random selection of study sites with *C. helmsii* from national distribution databases, no sites on peat soils were included. The avoidance of organic soil was apparent in the Bargerveen nature reserve, Province of Drenthe, the Netherlands (pers. obs.). In the periphery of this raised bog remnant, former farmland was restored through the removal of the nutrient-rich topsoil. There, *C. helmsii* was thriving where the mineral subsoil surfaced, but was absent in patches where peat remained.

Climate models predict higher winter temperatures at higher latitudes, drier summers and greater extremes in precipitation in the future [38]. This was illustrated by the dry growing seasons of the 2018–2020 period. These few growing seasons with little precipitation demonstrated that *C. helmsii* performs optimally only within a narrow range on the moisture gradient found on shores. On the wet side of the gradient (aquatic sites and recently exposed waterlogged terrestrial sites belonging to communities I, II and III), drought led to the expansion of *C. helmsii* into sites that were previously submerged and unfavorable due to carbon limitation. The desiccation of habitat made carbon dioxide in the atmosphere accessible to the plants. Unfortunately, a later inspection of some of these sites after new inundation in 2023 revealed that *C. helmsii* had not disappeared under the restored low $CO_2$ conditions and that its cover remained high, although the water levels returned to normal. Extremes in precipitation may thus provide *C. helmsii* with a window of opportunity to invade an environment that would otherwise be inaccessible for the species. Towards the dryer side of the gradient (communities VI and V), consecutive dry summers led to growth conditions that, half of the time, were too dry for the invader, resulting in the decline of *C. helmsii* cover. It is, however, unknown if these sites have an increased susceptibility to re-invasion or that the dry spell has permanently tipped the scales in favor of native plant species. Communities that did not change during the dry periods were found at the driest study sites where the *C. helmsii* cover remained low (community V) and at sites with a high and stable water level (community III). The latter were scarce, and the

absence of shores exposed to air depended on a continuous groundwater supply (Oud Hollandslaag), inlet of groundwater (Groot Meer) or steep banks (Molenheide).

The cover and biomass of invasive plant species can be limited through competition with native species (see Levine et al. [39] and the literature therein). For *C. helmsii*, this has been demonstrated in several experiments [7,15,16]. The growth of *C. helmsii* has experimentally been reduced through competition with *H. elodes*, *L. uniflora* and *P. globulifera*, as well as species introduced through *Ericetum tetralicis* clippings and a commercial grass/herb mixture suitable for moist conditions [15–17]. This field study confirms that *C. helmsii* will become the dominant species when no other dominant species are present. However, it also shows that the presence of a dominant native species does not guarantee a low cover of *C. helmsii* in the long run, at least not when weather patterns are unstable. Most of the native species that were abundant in 2016, and which might have been important resource competitors, disappeared in the four years of the study, and were replaced either by *C. helmsii* or by other native species that were less dependent on wet to moist conditions. Due to the interfering effect of fluctuating water levels, it is difficult to predict which native species are the most suitable competitors to *C. helmsii* when water levels are more stable. The only native species still present after four years were *L. uniflora* at aquatic sites and *J. effusus* at drier sites. These perennials have a thick cuticula, which enables them to withstand periods of drought.

### 4.2. Effect of C. helmsii on Native Flora

According to the European and Mediterranean Plant Protection Organization [40], *C. helmsii* is a major threat to native flora due to its rapid growth. However, various studies reported that the exclusion of native species does not occur when the species becomes dominant [41–44]. This study confirms the absence of an effect on species richness for aquatic sites, but not for terrestrial sites, where the richness correlated negatively with the cover of *C. helmsii*. Negative correlations between *C. helmsii* cover and the abundance of native species are frequently apparent ([41,44] and this study). However, negative correlations between *C. helmsii* cover and native species richness or abundance do not necessarily imply a causal relationship and an impact of *C. helmsii* on native species. Causality may even be reversed with native species, determining the performance of the invader, as various experimental studies have demonstrated strong resource competition between native species and *C. helmsii* [15–17].

Temporal data can provide better insight into *C. helmsii* dynamics and their possible impacts on native flora than one-time censuses. Therefore, we compared data from 2016 and 2020. Although *C. helmsii* is a notorious invasive species, the increase in cover was not a given, and the species increased as often as it decreased in cover. Would the plant community be affected by a change in *C. helmsii* cover, the expected impact would be low on species identity ($SER_r \approx 0$) due to a low species displacement [41–44], and high on species abundance resulting from the suppression of the species present ($SER_a \approx 1$) [41,44]. However, this was not the case, as turnover in the native community, both in terms of species identity and species abundance, was high independent of the change in *C. helmsii* cover. Furthermore, a change in *C. helmsii* cover did not affect the species richness.

Apparently, *C. helmsii* was not responsible for the high community turnover. The observed community turnover appears to be the result of the establishment and proliferation of species that are able to deal with the effects of a series of exceptionally dry years, rather than an increased cover of *C. helmsii*.

### 4.3. Limitations

In this study, relatively small plots were sampled in order to observe a clear relation between the cover and biomass of *C. helmsii* and local environmental conditions. However, there are also good arguments to sample on a larger scale to include other parameters, such as morphology, local hydrology and wind exposition. With the recent development of ortho imagery using drones, studies on larger spatial scales may become feasible over time. For

*C. helmsii*, the use of these techniques was recently evaluated by Strong [41]. He concluded that although the technology can be used to detect areas with a high cover of *C. helmsii*, the application is still limited; the species may be impossible to map when growing between other plants or when submerged.

In each study plot, only a single biomass sample was taken. In retrospect, pooling different samples would have yielded a more accurate biomass measurement, as it minimizes the effect of spatial variability.

Relationships between environmental parameters and the performance of *C. helmsii* were based on one-time measurements (i.e., 2016 in the Netherlands and 2019 in England). To distinguish between the biotic resistance of native flora and competitive displacement by *C. helmsii*, repeated measures are required. The repeated measures in this study were disrupted by an unexpected series of exceptionally dry years, which impacted the performance of *C. helmsii* as well as that of the native species.

The ability of this study to assess the possible impact of *C. helmsii* was further limited by the fact that half of the study sites had a high starting cover of *C. helmsii*. As a result, a further increase in cover was not possible at those sites. To assess invasion processes, it would have been better to seek out recently colonized sites where the cover of *C. helmsii* was still limited. That way, the opportunity for *C. helmsii* to expand at the study sites would not have been limited by its previous expansion but only by the environmental conditions present.

For the assessment of the impact of *C. helmsii* on native flora as well as the effectiveness of biotic resistance, we advise multi-year monitoring of sites differing in cover of native vegetation that have been recently colonized with a limited starting cover of *C. helmsii*.

This study is one of few that focuses on the variability of *C. helmsii* abundance under different environmental conditions [14] and the only one to analyze temporal variability outside of a controlled experimental setting [8,15–17]. In spite of its limitations, this study provided evidence that environmental parameters, which, according to experiments, are involved in the proliferation of *C. helmsii* ($CO_2$ [28]; N [15,16]), are also relevant under field conditions. Experimental studies have shown that various native species are able to control *C. helmsii* through resource competition [15–17]. However, this field study shows that the persistence of native competitors is limited to only a few species, probably due to a dominant effect of water shortage. The impact of *C. helmsii* on native flora also appears to be limited during subsequent dry years.

*4.4. Management of C. helmsii Invasions*

In the Netherlands, problems with an exuberant growth of *C. helmsii* often arise in water bodies with sparsely vegetated gently sloping banks. These have mineral soils, have slightly acidic to neutral water and are at least moderately rich in nutrients (N and C) ([15] and this study). These conditions are frequently found on former agricultural fields where the topsoil was removed in order to restore wetlands, moist habitat types, and endangered species [45–52], as well as in shallow soft water lakes and dune slacks, where measures have been taken to mitigate the effects of eutrophication and acidification [53,54]. Under these circumstances, *C. helmsii* prevents the recovery of native flora. In England, many of the study sites exhibited dominant *C. helmsii* populations in watercourses that received water from agricultural runoff that is rich in phosphate and inorganic carbon.

A number of measures can be taken at these sites to prevent the settlement and expansion of *C. helmsii* and thus protect the (endangered) native species that are present, as well as the investments that have been made to restore these sites. Measures consist of minimizing the nutrient load when restoration projects are being carried out, as well as reducing the input of nutrients through surface water, atmospheric deposition, waterfowl and other sources afterwards. According to Brouwer et al. [15], nitrogen as well as phosphorus are important nutrients that can stimulate the growth of *C. helmsii* on mineral soils. This study demonstrates that it is also important to take inorganic carbon into account as an important nutrient in submersed situations.

Where possible, steps should be taken to reduce water level fluctuations in order to minimize the area with optimal moisture for *C. helmsii*, as well as to prevent or minimize the expansion of the species into the lower parts of wetlands during dry periods. A more stable hydrology should also provide native species with stable growth conditions, reducing turnover within native species assemblages and thus providing a smaller window of opportunity for *C. helmsii*. This advice appears to contrast with the suggestion of Brouwer et al. [15] to induce periods of superficial desiccation after *C. helmsii* appearance to prolong the phase of limited expansion. However, a period of desiccation may provide time to eliminate the species or take other actions to prevent its expansion (such as introduction of competing species), and, as such, is a temporary measure. Creating wetter conditions and fewer periods of desiccation by optimizing hydrology, on the other hand, is a long-term measure that maximizes carbon limitation in the water to prevent the expansion of *C. helmsii*.

A variety of native species have been shown to compete sufficiently with *C. helmsii* to reduce its growth, such as the previously mentioned *H. elodes*, *L. uniflora* and *P. globulifera*, and species from moist heath- and grassland [12,15–17]. This study identified a total of 24 native species (Table 4) that can dominate when the *C. helmsii* cover is low and can be considered for introduction in order to increase biotic resistance. However, in order to anticipate varying environmental conditions, one should prioritize species with a broad niche. *L. uniflora* is the only species with a stable presence in the communities with fluctuating water levels that performs well under nutrient-poor conditions on shores as well as under water [55]. *J. effusus*, also a stable dominant species, is a good choice for nutrient-rich shores [56]. Another way to deal with environmental change is to add redundancy by introducing multiple competing species with comparable functional traits [10]. As different species are likely to respond differently to environmental change, the chances increase that one or more will survive and maintain biotic resistance to *C. helmsii* invasion. Furthermore, obtaining a higher species richness with the introduction of species with similar functional traits can maximize niche overlap, providing better biotic resistance [39,57,58].

Instead of preventing the settlement and expansion of *C. helmsii*, the above-mentioned methods can also be used to restore wetlands that have already been invaded. This will require an effort to reduce the cover and biomass of *C. helmsii* [59] before measures are taken for nutrient reduction, hydrological restoration and the introduction of competing species. Many methods have been tested, which may have been unsuitable for the elimination of *C. helmsii*, but are adequate for a temporary population reduction [6,44]. Although some practical challenges remain, such as the reduction in *C. helmsii* cover under water and the acquisition of sufficient propagules of native species, the first trials focusing on the management of *C. helmsii* populations through manipulating biotic and abiotic conditions are promising [17].

With the increasing number of invasive species, active population control is becoming more difficult and costly, and the need for a more nature-based solution is increasing. Optimizing ecological filters and stimulating competing native species is such a solution for which field and experimental studies have been gathering evidence. It is time to take the next step and put this theory into practice.

**Supplementary Materials:** The following supporting information can be downloaded at https://www.mdpi.com/article/10.3390/d16030172/s1, Table S1: The names and coordinates of the study sites.

**Author Contributions:** Conceptualization, H.H.v.K.; data curation, H.H.v.K. and J.M.M.v.d.L.; formal analysis, J.M.M.v.d.L. and L.S.v.V.; funding acquisition, H.H.v.K.; methodology, H.H.v.K.; visualization, H.H.v.K.; writing—original draft, H.H.v.K., J.M.M.v.d.L. and L.S.v.V.; writing—review and editing, H.H.v.K. All authors have read and agreed to the published version of the manuscript.

**Funding:** This study was financially supported by the subsidy 'Biodiversiteit en Leefgebieden' [grant code C2165546/3825182] from the Province of Noord-Brabant, the Netherlands; a grant from the NVWA order number 60005870; a grant from the OBN program, financed by the Dutch Ministry of Agriculture, Nature and Food Quality and BIJ12; and a grant from Stichting Ontwikkel-en Innovatiecentrum Bargerveen.

**Data Availability Statement:** The data presented in this study are openly available in the Harvard Dataverse at https://doi.org/10.7910/DVN/WUXLEQ (accessed on 1 March 2024).

**Acknowledgments:** We are grateful to Natuurmonumenten, Brabants Landschap, Staatsbosbeheer and Unie van Bosgroepen for allowing us to conduct field studies and sampling. The authors would also like to thank the staff of Bargerveen Foundation and Radboud University for their advice and technical assistance. Special thanks to Conor Strong for his help during sampling in England, and to Paul van Els for improving the English in the manuscript. We thank the editor and two anonymous reviewers for their constructive comments on the manuscript.

**Conflicts of Interest:** The authors declare no conflicts of interest. The funders had no role in the design of the study; in the collection, analyses, or interpretation of data; in the writing of the manuscript; or in the decision to publish the results.

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
