# Peer review of "Low Resource Competition, Availability of Nutrients and Water Level Fluctuations Facilitate Invasions of Australian Swamp Stonecrop (Crassula helmsii)"

_diversity, doi:10.3390/d16030172_

Round 1

Reviewer 1 Report

Comments and Suggestions for Authors

I recommend changing the title since the results from the paper are adding to the understanding of the ecology of C. helmsii, but not its understanding. Something in the line of “Insights from field data …” or even better, highlight your major result in the title.

The Abstract has room for improvement, especially in its introductory section.

Introduction

More emphasis should be made on the problematic with this invasive species. Why is it so important to manage it?

Line 43-44: make the sentence clearer with the use of the active voice. The sentence is not clear… everywhere which species grows? Is this observation from the authors? If not, there is a reference missing. Please clarify.

Line 47: it looks like the description of the conditions under which the species thrives is pretty good for this species. Thus, why is it anecdotal?

Line 68-71: revise the English

Line 68: What is the aim of the study? Which are the hypotheses? Why is it important or needed? It looks to me that having information in Dutchland is one of the reasons as the invasiveness/invasive behavior there might be different from other places in the world.

Methods

Line 84: how many water samples per site? Please specify

Line 94: how did you sample the soil? If it was underwater, how did you do it? Please specify

Line 106: Why did you only sampled one 0.25m2 quadrat per site for biomass? It would have been better to sample 3 quadrats randomly (maybe with a smaller quadrat) to account for spatial variability.

Line 108: cover was visually estimated? Do the observers were previously trained in visually estimating plant cover? Did you use the Braun-Blanquet survey methodology? Please specify.
It would have been good to estimate cover thanks to orthoimagery from drones for example, since they are readily available nowadays. The cover estimation would have been more representative, since you would have covered more area than 16 m2. You could include this in your recommendations.

Line 147: erase ‘of’ from sentence

clarify when sampling was done. It appears in the results that sampling was continuous from 2016-2021. If this was the case, it should be clearly stated in the methods as well as the month of sampling, that could also influence the plant community present.

          One of the key aspects of your dataset compared to published literature is that you have field data for many sites (48) and for some years (6 years if I understood well) with plant community composition. Your dataset is valuable and can bring light in the matter of the effects of the invasive species in the loss of biodiversity. Thus, I would recommend taking advantage of this to explore biodiversity changes using a more advanced indicator of biodiversity change than species richness. The use of richness-based species-exchange ratio is more appropriate and, even better, the use of an indicator to assess differences between species proportional abundances as a measure of turnover by changes in species proportional abundances. See this paper for the justification and formulas: Hillebrand, H., Blasius, B., Borer, E. T., Chase, J. M., Downing, J. A., Eriksson, B. K., et al. (2018). Biodiversity change is uncoupled from species richness trends: Consequences for conservation and monitoring. Journal of Applied Ecology 55, 169–184. doi: 10.1111/1365-2664.12959

Results :

Results should be accompanied by p-values and statistical values, or a reference to Table 1 provided in the text where appropriate.

Figure 1 is not very helpful as it is expected that as plant cover increases so does the biomass. What is interesting is the difference between terrestrial and aquatic growth. You can differentiate with different colours the terrestrial and aquatic plants in your graph.

Line 244: you introduce sampling in 2020 while in the Methods sampling seemed to have occurred in 2016, 2019 and 2021 at the Dutch sites. Please correct.

Table 3 is not understandable.

You should provide information on the rainfall for each studied year since it appears to be an explaining factor affecting C. helmsii invasiveness.

Discussion

C. helmsii appears to be a fierce competitor for native flora, reduces the germination of native plants and even suppress native species within few years of introduction (see https://www.iucngisd.org/gisd/speciesname/Crassula+helmsii and references therein). However, your results seem to suggest that C. helmsii is not so good competing with native flora, and that it is not even the dominant species at many sites. You demonstrate that in aquatic sites, richness is not affected by C. helmsii, however in terrestrial sites, there are statistical differences in terms of richness.

Line 288: you need to provide a reference

Line 292: was this measured at each site? If so, it should be presented in the Results. If not measured by you, then it should be referenced.

Line 299: you need to provide a reference

Line 307: you need to provide a reference

Line 361: You need to tune down your affirmation since the results from this study seem to suggest that although it does not prove it was the variability in water levels and resource competition.

Line 369: revise the English and include reference(s)

Data Availability Statement: the link appears to be wrong

Supplementary 2: the synoptic table provides numbers for all years and all sites confounded? This is of little utility since your sites cover a wide range of places and years.

Clearly delineate the context-specific limitations of your study and discuss how the study contributes to the broader field.

I encourage you to address these concerns comprehensively in a revised manuscript. Emphasize the potential benefits of your study, discuss its limitations, and suggest avenues for future development. A deeper analysis, along with improved readability, will greatly enhance the overall quality of the paper.

Comments on the Quality of English Language

The manuscript should be thoroughly revised to improve the quality of English language.

Reviewer 2 Report

Comments and Suggestions for Authors

The abundance of C. helmsii in northern Europe has been studied in relation to nutrient availability, water availability and the presence of coexisting plant communities and plant species.

Comments:

The Methods section should be clear and detailed enough for another experienced person to repeat the research.

L 90-91 How concentrations of CO2 and HCO3 were determined?

L 96-98 Sequence of work: diluted, then dried?

The description of the determination of total P in soil samples is insufficient

L 113 What size subsample was? How was biomass calculated in this and other cases?

L 135 How is abundance measured? One can only guess that it is cover in percentages, but precision is needed.

L 138-141 Very vague description.

Equation 2 and its use is unclear.

How data on expected biomass was obtained? 

L 147 What dataset?

L 148 How can a plant be called non-invasive in an area where it covers, e.g., 40%?

L 150 To which group are assigned the sites with mean EV 3-4?

L 152 How cover richness was measured.

How was alkalinity measured?

Two groups of sites were compared in terms of C. helmsii abundance. Why the correlation between the abundance of these plants and the water and soil chemistry was not assessed using data from all sites together?

Results

L 159 What is gr and DW?

Figure 1 Title and titles of axis need correcting. Coverage is an alternative American usage. Units of measurement are duplicated

L 170 Does C. helmsii grow submerged underwater in aquatic sites?

L 172-174 The sentence is completely unclear.

L 183 pH 8.5 not neutral.

Table 1 p > 0.0001 Are you sure you are using the right sign? Mean is not a unit of measurement. Think about a more rational table layout.

Second Figure 1 Title should be changed into more appropriate. What is actually represented on the ordinate axis in the figure, since under no circumstances, the standard deviation (SD) can be negative or less than zero.

L 215 Weren't all the sites studied of these communities invaded?

Table 2. The top rows must be rearranged. In which year were the studies carried out?

L 236-237 It is not clear. Has C. helmsii moved to another community or has the community changed? In the second case, C. helmsii could not have shifted communities because it is not a characteristic species.

L 238 What do you mean by invaded? Was C. helmsii absent in the other sites?

L 242-243 Cover of C. helmsii in community V is the largest (84.4%).

Table 3 is unclear and needs more explanation.

Table 4. The 4 top rows must be rearranged. Really, no dominant species? Wasn't C. helmsii the dominant species there?

L 259 ‘C. helmsii increased‘?

L 274-275 ‚... growth conditions, such as algae...‘ ?

Discussion

L 282 What really depends on what?

L 306 Not explained abbreviation.

L 307-309 Where does this information come from? But in dense stands CO2 was also 48.4

L 333 Is communities I, II and III wet or aquatic?

L 364 J. effuses

L 383, 393 Brouwer et al.

Round 2

Reviewer 1 Report

Comments and Suggestions for Authors

Thanks for providing the revised version of the document. Most of the concerns have been addressed, and the manuscript looks a lot better now. However, there are still some further work needed to improve the work:

Line 216: it looks like the word “met” is missing in the sentence (were not comparisons between)

Line 219: R should be referenced (see R documentation). If you used any package, you should also state it and provide the reference.

Line 301: include the word significantly, since the statistics are showing a statistically significant change

Figure 1: I still consider that this graph can be improved. Now that we see the differences between aquatic and terrestrial, I can see there are actually not so many differences… apparently.
What I can see is that there is an asymptote to which the biomass saturates with cover. Please consider adding a smooth line to the data (with its CI) to help visualize this tendency.

Figure 3: I appreciate these new figures showing the relationship between SERr and SERa with change in C. helmsii cover, however I am not really sure the regression line and the (non-significant) correlation is showing the real distribution of the data.

·        Figure 3A there is no pattern at all, richness is high and low independently on the changes on C.h cover. Meaning = C.h cover change does not affect vegetation richness

·        Figure 3B SERr is higher at low and high C.h cover change, while when C.h does not change, then SERr has low, middle and high values
Meaning: species turnover is high when C.h cover change is negative and positive.

·        Figure 3C similar to 3B although much less pronounced.

If you want to see a trend in this data, I suggest using a GAM or similar to really show what is happening with the data. I believe it is not correct to summarize the trend with a line in this case.

Please adjust the interpretation of your results for species change, species turnover, and abundance turnover in the Results and Discussion sections accordingly.

Table 3: further improvement is needed; you can combine the first column with the 2016 number in horizontal orientation. What do the numbers mean?

Regarding this comment, I cannot see any change in the document at the place you point me to (Line 439-462):

Comment: C. helmsii appears to be a fierce competitor for native flora, reduces the germination of native plants and even suppress native species within few years of introduction (see https://www.iucngisd.org/gisd/speciesname/Crassula+helmsii and references therein). However, your results seem to suggest that C. helmsii is not so good competing with native flora, and that it is not even the dominant species at many sites. You demonstrate that in aquatic sites, richness is not affected by C. helmsii, however in terrestrial sites, there are statistical differences in terms of richness.

 Response: We added a paragraph to the discussion section describing the (lack of) impact of C. helmsii on biodiversity metrics. Line 439-462

Finally, regarding the Data Availability Statement: the DOI still doesn't work and doesn't lead to the data... please provide a working link for this.

Comments on the Quality of English Language

All English language concerns are embedded in the previous comments.
